# The Effect of Social Network Use on Chinese College Students’ Conspicuous Consumption: A Moderated Mediation Model

**DOI:** 10.3390/bs13090732

**Published:** 2023-09-01

**Authors:** Lei Xu, Zhaoxizi Lu, Lingyun Wang, Jiwen Chen, Lan Tian, Shuangshuang Cai, Shun Peng

**Affiliations:** School of Education, Jianghan University, Wuhan 430056, China; xulei@jhun.edu.cn (L.X.); lzxz0518@163.com (Z.L.); speng2psy@gmail.com (L.W.); chenjiwen@jhun.edu.cn (J.C.); duncantian@163.com (L.T.); zhumodanqing@126.com (S.C.)

**Keywords:** use of social networking sites, conspicuous consumption, upward social comparison, optimism, college students

## Abstract

This study explored the effects of social networking site use intensity, upward social comparison, and optimism on college students’ conspicuous consumption and their mechanisms of action using a sample of Chinese college students. A total of 717 Chinese college students (M = 20.08, SD = 1.44; 73.9% female) completed the Social Network Use Intensity Scale, the Upward Social Comparison Scale, the Life Orientation Test, and the Conspicuous Consumption Scale. The results indicate that (1) the intensity of use of social networking sites significantly positively predicts the conspicuous consumption behavior of college students; (2) upward social comparison plays a mediating role between the intensity of social networking site usage and conspicuous consumption; and (3) optimism moderates the second half of the mediating path between the intensity of social networking site use, upward social comparison, and conspicuous consumption. Specifically, the relationship between upward social comparison and conspicuous consumption among college students with low optimism levels is stronger than that among college students with high levels of optimism. Intensity has a stronger positive effect on conspicuous consumption through upward social comparison. It is concluded that the intensity of college students’ use of social networking sites can affect their conspicuous consumption behavior through upward social comparison, and this relationship is moderated by optimism. The results of the study help to reveal the influence of SNS (social networking site) use behavior on conspicuous consumption and its mechanism of action and have implications for reducing the negative impact of conspicuous consumption on college students.

## 1. Introduction

According to the 2021 Government Work Report, China has made historic achievements in economic and social development, with its GDP exceeding CNY 100 trillion [1]. In this context, people no longer focus only on the practical value of commodities in consumption but pay increasing attention to the psychological added value produced by commodities. With the rise of modern consumerism, people’s concepts of consumption have changed. Tmall Luxury Bain & Company released the “China Luxury Market 2020: Unstoppable” China Luxury Consumption Survey Report in 2020, which shows that the website transaction amount increased by approximately 150% compared to the previous year and that the college student group is rising as an emerging luxury consumer [2]. This indicates that the demand of college students for luxury goods is increasing and reflects the changing consumption characteristics of the college student consumer group. Under these circumstances, the phenomenon of conspicuous consumption has become a major trend. Conspicuous consumption refers to the behavior of individuals who reinforce their image by consuming openly or showing their status to others [3]. It is specifically manifested when individuals seek the nonfunctional value of goods to purchase goods or services that are symbolic and able to demonstrate their status or build their self-image to others [4]. As a kind of irrational consumption, conspicuous consumption not only tends to cause an economic burden to individuals but also may form wrong and distorted consumption views and values, which affect individuals’ physical and mental health [5].

Additionally, the unique Chinese culture of “mianzi” has turned this phenomenon into a growing trend [6,7]. “Mianzi” refers to an individual’s success in gaining his own recognition and the positive social value that others believe he deserves [8] and is an important consumer motivation that distinguishes Chinese consumers from those in other countries. Under the influence of collectivism, the Chinese ego places more emphasis on the individual’s relationship with others and treats close others as an important part of the self [9]. This awareness of face can cause individuals to place more emphasis on the external characteristics of the good, such as whether it accentuates status, rather than the intrinsic characteristics of the good itself, such as the use value and quality, when consuming it [10]. Compared to consumers in the United States, Chinese consumers are more likely to be influenced by reference groups as well as prestige motives [11]. For Asians, face and group orientation influence the conspicuousness dimension of their luxury consumption [12]. Consequently, conspicuous consumption in the context of Chinese culture may be more prevalent. Researchers also note that conspicuous consumption is becoming more common among college students [13,14]. However, the conspicuous consumption behavior of college students may lead to more serious negative consequences, such as deviant behavior, school loans, and even “naked loans” [15]. From this perspective, there is an urgent need to discuss the mechanism of the conspicuous consumption behavior of college students.

## 2. Literature Review

Currently, domestic and foreign scholars’ research on the mechanism of conspicuous consumption mostly focuses on factors in the traditional environment [16], namely, status [17,18], power [19], gender [20], courtship [21], income [22], materialism [23], and other factors. At present, compared with the offline environment, college students share their personal lives more often through social networking sites such as WeChat, QQ, and Weibo. On the one hand, social networking sites provide various ways to show self-presentation, such as posting text and uploading photos and short videos. On the other hand, many users pay attention to self-presentation, and self-presentation social networking sites have become a new form of self-presentation for college students. Conspicuous consumption is essential to presenting oneself to others. As one of the forms of self-presentation in the field of consumption, it is particularly necessary to study the mechanism of the influence of the online context on college students’ conspicuous consumption.

However, there is a relative lack of research on the factors influencing online contexts, and only a small number of scholars have studied and confirmed the positive predictive effect of SNS (social networking site) use on conspicuous consumption [24,25,26,27]. Social network sites (SNSs) are web services based on a bounded system, such as WeChat, Weibo, and QQ. Through public or semipublic personal pages, users are able to connect with each other and view and forward links to themselves as well as to other users and use social networking sites to present themselves or to develop and maintain relationships with others [28,29]. Due to their fast information dissemination and wide range of dissemination, social networking sites have become an important platform for interpersonal communication in the information age. As of June 2021, the usage rates of QQ and WeChat, the two major social media platforms in China, reached 60.6% and 88.3%, respectively [30], and online social networking has been deeply integrated into people’s daily lives. Studies have found that the use of social networking sites can promote self-esteem and life satisfaction in young people [31], but the excessive use of social networking sites may lead to negative effects on individuals, such as stress, negative emotional experiences, and decreased quality of life and health [32,33,34]. While previous studies have addressed the psychological effects of SNS use on individuals, fewer have explored the effects of SNS use on individual behaviors (e.g., consumption).

Veblen (2019) pointed out that social networking influences consumption and that the information shared by many users on social networking sites may trigger conspicuous consumption. Currently, the function of online social networking is not limited to communication, but social networking sites themselves are one of the ways to show off. On the one hand, the content shared on social platforms contains personal shopping experiences as well as reviews about products, and receiving goods containing symbolic information such as luxury goods may have an impact on users’ consumption attitudes and behaviors [35]; on the other hand, other users met on social networking sites, as an extension of real-life interpersonal relationships, may also influence individuals’ purchase decisions [36]. With the use of social networking sites, individuals can not only have a more comprehensive understanding of the lives of others but also imperceptibly change their concept of consumption and engage in conspicuous consumption with more personalized and distinctive characteristics. Subsequent researchers have shown that the excessive use of social networking sites may reduce people’s self-control, leading to more impulsive or indulgent behaviors and increasing people’s spending on conspicuous goods [24,25,27]. For example, Taylor and Strutton (2016) found that the use of social networking sites can positively predict conspicuous consumption [37]. This study intends to further explore the relationship between SNS use and conspicuous consumption and its internal mechanisms. Thus, we can formulate the following hypothesis:

**H1.** *The intensity of SNS use significantly and positively predicts college students’ conspicuous consumption behavior*.

First, the mediating mechanisms of social networking site use regarding conspicuous consumption have been less discussed in previous studies. Researchers have recently argued that it is not enough to explore direct links between variables but that potential mediating variables need to be introduced to answer the question “How does social networking site use act on conspicuous consumption?” It has been suggested that social comparison on social networking sites may be an important factor [38]. Individuals’ use of social networking sites leads to social comparison by viewing information about others’ self-presentation and interactions with others, and the intensity of social networking site use is positively related to social comparison [39]. Compared with traditional social activities, social comparisons on social networking sites are characterized by spontaneity, the polarization of comparative information, and the diversity of comparative objects [40]. As most of the posting dynamics on social networking sites have impression management, users of the sites tend to present the positive and idealized side of their lives, and this information may lead other users to believe that others are in a better state of life than they are [41], and social comparisons resulting from social networking sites are more likely to be upward [42,43]. Upward social comparison refers to individuals’ unconscious comparison of themselves with those who are better than themselves in the process of using social networking sites [44]. Research has shown that people with a higher intensity of social media use are exposed to superior information about others for longer periods, are more likely to make upward social comparisons, and, as a result, have a range of negative emotions [45]. In the process of using social networking, people tend to release positive information that is beneficial to shaping their self-presentation [46], such as beauty selfies, carefully arranged breakfasts, and beautiful scenery. When individuals browse this information, they unconsciously compare it with their own situation to better self-evaluate, and the presentation of this positive or idealized information induces upward social comparison [47,48,49]. In addition, under the influence of upward social comparison, individuals often develop a sense of psychological disadvantages, such as “I am not as good as others”, which leads to lower self-evaluation [49], and the more extreme the upward social comparison is, the more likely it is to lead to a sharp decline in self-evaluation [38]. The self-evaluation maintenance model [50] also suggests that individuals engage in social comparisons to obtain positive self-evaluations. When individuals perceive a gap between their current standard of living and that of others during upward social comparison and their self-evaluation is threatened, they may increase the likelihood of conspicuous consumption to maintain a positive self-evaluation [51,52]. Therefore, the following can be assumed:

**H2.** *Upward social comparison plays a mediating role between the intensity of social network use and conspicuous consumption*.

Second, existing studies have explored the moderating mechanisms by which SNS use affects conspicuous consumption less. Although social networking site use affects conspicuous consumption either directly or indirectly, there are individual differences in this influence mechanism, so it is necessary to examine whether the relationship between social networking use and conspicuous consumption is moderated by other factors. The selective access model [53] suggests that when practicing social comparison, individuals process information about the difference between themselves and the comparison target, which can produce two completely different results [54]. The first is the contrast effect, that is, there is a large difference between oneself and the comparison target (in the context of upward social comparison, the level of self-evaluation will be weakened). The other is the assimilation effect caused by the similarity of the comparison goal (self-evaluation is improved in the face of upward social comparison). As a stable personality trait, optimism is considered part of positive psychological capital, which means that individuals have positive psychological expectations for the future [55] that affect the processing of social comparison information. In particular, Collins’ upward assimilation theory [56] suggests that individuals’ perception of their own future achievement status affects the results of upward social comparison. Specifically, when people receive upward social comparison information about an ideal life or a positive self-image from the outside world, individuals with high levels of optimism may have a psychological expectation that they can achieve the same level in the future based on positive cognition and their evaluation of themselves and the external environment. This kind of expectation weakens the contrast effect or produces the assimilation effect, alleviates the negative impact of upward social comparison, and makes conspicuous consumption less likely.

In contrast, individuals with a low level of optimism will be more sensitive, more susceptible to the negative impact of upward comparison information, and more likely to damage their self-concept. According to psychological compensation theory [18], individuals may compensate for a certain aspect of their psychological deficit through other means. Individuals with a low level of optimism may compensate for the self-threatening effects of lower self-evaluation by choosing conspicuous goods that demonstrate their value when they are threatened by upward social comparison. Thus, the following is postulated:

**H3.** *Optimism moderates the second half of the path by which the intensity of social networking site usage affects conspicuous consumption through upward social comparison. In other words, upward social comparison has a greater predictive effect on conspicuous consumption among college students with a low level of optimism than among college students with a high level of optimism*.

Therefore, this study intends to explore the relationship between the use of social networking sites and conspicuous consumption among college students and to examine the mediating and moderating effects of upward social comparison and optimism.

## 3. Materials and Methods

### 3.1. Participants

In this study, college students from five universities in mainland China were selected as the subjects, and the data were collected through an online survey using a whole-group random sampling method. The questionnaires were distributed with the help of counselors at each university. The online questionnaires were distributed uniformly by the counselors to the students of the grade for which they were responsible. The researcher used strict control measures to ensure the authenticity of the data, such as checking for the presence of duplicate IP addresses and the length of time students spent filling out the questionnaires. Based on reviewing the quality of all questionnaires, invalid questionnaires, such as blank questionnaires, questionnaires with large missing data, and questionnaires with regular responses, were excluded. There were 814 study participants, and 717 valid data points were recovered (valid recovery rate of 88.08%). Valid participants were aged 17–26 years (*M* = 20.08, *SD* = 1.44), of whom 530 (73.9%) were female. All students were informed before completing the questionnaire that participation in this survey was completely voluntary, that there would be no penalty for not participating, and that the study data were reported as group results with no personal information involved. All subjects who participated in the study signed an informed consent form. This study was approved by the Ethics Committee of Jianghan University.

### 3.2. Measures

#### 3.2.1. Social Network Use Intensity Scale

In this study, the Social Network Use Intensity Scale developed by Ellison et al. [31] and revised by Niu et al. (2016) was used [57]. The questionnaire has eight items. The first two items measure the number of friends and the average daily usage time of an individual on social networking sites, and the last six items are scored on a five-point Likert scale (one “disagree very much” to five “agree very much”), which measures the strength of the emotional connection between social networking sites and individuals and the degree to which these sites penetrate individual lives. After converting individual scores into standard scores, the total mean score is calculated. The higher the score is, the higher the intensity of social networking site use. The Cronbach’s alpha of this questionnaire in this study was 82. Confirmatory factor analysis showed that the model had good construct validity (*χ^2^* = 182.53, df = 20, CFI = 0.90, TLI = 0.87, SRMR = 0.05).

#### 3.2.2. Upward Social Comparison Scale

This study used the Upward Social Comparison Scale, a subscale of the Comparison Tendency Scale, which was translated by Bai et al. (2013) [58] and compiled by Gibbons and Buunk [59]. The questionnaire contains six items that are scored on a five-point Likert scale (one “disagree very much” to five “agree very much”). The higher the score is, the greater the tendency of individuals to make upward comparisons in their use of social networking sites. The Cronbach’s alpha for this scale in this study was 91. Confirmatory factor analysis showed that the model had good construct validity (*χ^2^* = 207.59, df = 9, CFI = 0.93, TLI = 0.88, SRMR = 0.05).

#### 3.2.3. Life Orientation Test

This study used the Life Orientation Test (lot-R) developed by Liu and Cheng (2007) [60] to measure the optimism level of the participants. The questionnaire contains 12 items, with 5 items for optimism and pessimism; the rest are interference items, which are not included in the total score. The test is scored on a five-point Likert scale (one “disagree very much” to five “agree very much”). This study refers to Liu et al. (2017) [49]. By reverse-scoring the data of pessimistic items, the sum of the scores after the conversion of the optimistic dimension and the pessimistic dimension predicts the level of individual optimism. The higher the score is, the higher the level of individual optimism. The Cronbach’s alpha of this questionnaire in this study was 82. Confirmatory factor analysis showed that the model had good construct validity (*χ^2^* = 89.53, df = 22, CFI = 0.97, TLI = 0.95, SRMR = 0.04).

#### 3.2.4. Conspicuous Consumption Scale

This study used the Conspicuous Consumption Scale compiled by Marcoux [24] and revised by Chen (2009) [6]. The questionnaire has 13 items and is scored on a five-point Likert scale (one “disagree very much” to five “agree very much”) to measure an individual’s conspicuous consumption tendency. The higher the score is, the stronger the conspicuous consumption tendency. The Cronbach’s alpha for this scale in this study was 91. Confirmatory factor analysis showed that the model had good construct validity (*χ^2^* = 395.49, df = 59, CFI = 0.94, TLI = 0.92, SRMR = 0.05).

### 3.3. Procedure and Data Processing

This study used SPSS and Hayes’ (2013) [61] SPSS macro program PROCESS to organize and analyze the data. This SPSS macro program can be used to validate a wide range of mediated moderation and mediated models with moderation based on the bootstrap method of bias correction percentile and has been used by others to test whether the first or second half of the mediating effect is moderated [62,63].

In this study, descriptive statistics, correlation analysis, and common method bias tests were performed on the data using SPSS. Model 4 of the SPSS PROCESS macro was used to test the mediating effect model, and model 14 of the SPSS PROCESS macro was used to test the mediating effect model with moderation. The bootstrap method was used to sample 5000 times in SPSS PROCESS macro, and the percentile bootstrap was used to estimate the mediating effects and their confidence intervals.

## 4. Results

### 4.1. Common Method Deviation Test

To avoid common method deviation, the corresponding control was carried out in the program when collecting data, such as anonymous answers and reverse-scoring of some items. After data collection, the Harman single-factor test [64] was used to detect whether there was a common method bias. The results showed that there were eight extracted factors (eigenvalues > one), among which the variation explained by the first factor was 20.69%. Therefore, there was no serious common method bias.

### 4.2. Descriptive Statistics and Correlation Analysis of Each Variable

As shown in Table 1, the intensity of SNS (social networking site) use is significantly positively correlated with upward social comparison, conspicuous consumption, and optimism. Conspicuous consumption is positively correlated with upward social comparison. Upward social comparison is negatively correlated with optimism. Conspicuous consumption is not significantly associated with optimism.

### 4.3. Mediation Model Testing

After standardizing the independent variable social networking site use, the dependent variable conspicuous consumption, and the mediating variable upward social comparison, the mediating role of upward social comparison between social networking site use and conspicuous consumption was analyzed using Hayes’ SPSS macro program PROCESS.

First, this study used Model 4 in the SPSS macro program compiled by Hayes (Model 4 is a simple mediation model) to test the mediating effect of upward social comparison between the intensity of social network use and conspicuous consumption. The results showed that the intensity of SNS use could significantly predict upward social comparison (β = 0.26, *SE* = 0.05, *p* < 0.001) after setting age and gender (dummy variable 0 for males and 1 for females) as control variables. When upward social comparison was included, that is, the intensity of social network use and upward social comparison were both entered into the regression equation, the intensity of social network use significantly predicted conspicuous consumption (β = 0.19, *SE* = 0.04, *p* < 0.001). Upward social comparison also significantly predicted conspicuous consumption (β = 0.26, *SE* = 0.03, *p* < 0.001). The bootstrap test results showed that upward social comparison had a significant mediating effect between the intensity of social network use and conspicuous consumption (0.07, 95% confidence interval (0.03, 0.10)). The mediating effect accounted for 26.51% of the total effect.

### 4.4. Moderated Mediation Model Testing

Model 14 was used to test the moderating effect of optimism and to analyze whether the mediating role of upward social comparisons between social networking site use and conspicuous consumption (the second half) is moderated by optimism. To judge whether a moderated mediation effect exists, the following four conditions need to be met: (a) in Equation (1), the intensity of social network use has a significant predictive effect on conspicuous consumption; (b) in Equation (2), the intensity of social network use has a significant predictive effect on upward social comparison; (c) in Equation (3), the main effect of upward social comparison and conspicuous consumption is significant, and the interaction effect of optimism and upward social comparison is significant; and (d) after the interaction term enters the regression equation, *R*^2^ is greater than the explanatory power of the model without the interaction term. The results presented in Table 2 show that Equation (1) is significant and that the intensity of social network use positively predicts conspicuous consumption, satisfying condition (a). In Equation (2), significantly, the intensity of SNS use is positively predicted by upward social comparison, satisfying condition (b). Equation (3) is significant. Upward social comparison positively predicts conspicuous consumption, and the interaction term between optimism and upward social comparison is significant, satisfying condition (c). In Equation (3), *R*^2^ is 0.17, which is greater than 0.04 in Equation (2) without the interaction term, which satisfies condition (d).

To further analyze the specific impact of the regulatory effect, a further simple effect analysis was conducted. The results are shown in Table 3. According to the optimism scores, those greater than one standard deviation from the mean were divided into the high optimism group, and those less than one standard deviation from the mean were divided into the low optimism group. The results showed that (see Figure 1) for college students in the low optimism level group, upward social comparison was a significant positive predictor of conspicuous consumption (*β*_simple_ = 0.33, *p* < 0.001), while for college students in the high optimism group, the positive predictive effect of upward social comparison on conspicuous consumption was weakened (*β*_simple_ = 0.19, *p* < 0.001). Overall, the indirect effect of social network use intensity on conspicuous consumption through upward social comparison is moderated by optimism. For college students with a high optimism level, the indirect effect index = 0.09 and boot SE = 0.03, with a 95% confidence interval (0.04, 0.14); for college students in the low optimism group, the effect index = 0.05 and boot *SE* = 0.02, with a 95% confidence interval (0.02, 0.08).

## 5. Discussion

Based on the current era of information society and the actual life of college students, this study explores the influence of SNS use on college students’ conspicuous consumption and its mechanism of action and analyzes the mediating role of upward social comparison and the moderating role of optimism. The results of the study help deepen the understanding of the relationship between SNS use and conspicuous consumption and its internal mechanism of action.

The results of the study showed that the intensity of social networking site use was a significant positive predictor of conspicuous consumption, and the higher the intensity of social networking site use among college students, the more likely they were to engage in conspicuous consumption, in line with hypothesis H1. This verifies the findings of previous studies, indicating that social networking site use is significantly and positively related to conspicuous consumption [35]. Individuals spend most of their time browsing information on social networking sites. Most of the information shared by users of social networking sites is embedded with commodities containing symbolic value, thus potentially influencing the attitudes and behaviors of visitors [35]. Other studies have shown that the use of social networking sites induces individuals’ negative emotional experiences [31]. In this context, consumption becomes a way for individuals to compensate for psychological damage. In other words, the people and things that people come into contact with during the high-intensity use of social networking sites increase the possibility of conspicuous consumption.

On this basis, this study further found that upward social comparison plays a mediating role in the relationship between the intensity of social network use and conspicuous consumption. In other words, the intensity of social network use not only has a direct positive prediction effect on conspicuous consumption but can also affect conspicuous consumption through the intermediary effect of upward social comparison, which verifies hypothesis H2. This finding can be explained from the following perspectives. First, college students are an important part of social comparison on social networking sites, where they spend much time and energy on social interactions with friends, such as browsing and evaluating information posted by others. It is believed that when individuals are exposed to others’ information, they subconsciously relate others’ information to themselves, and the presence of others’ information can trigger individuals’ social comparisons [65]. Social networking sites provide a platform for people to reach more users than traditional face-to-face communication, and many posts and messages provide individuals with more opportunities for social comparison, which automatically occurs when viewing content about others’ achievements [66,67]. Thus, individuals who use social networking sites more frequently are more likely to be exposed to the information posted by other users compared to individuals who use social networking sites less frequently, which inadvertently provides more possibilities for social comparison.

In addition, the anonymity and convenience of online social networking allow people to selectively present their personal images and lifestyles, giving them more control over impression management, and users of social networking sites tend to selectively present positive aspects of themselves [68], thereby shaping others’ positive impressions of them. When individuals use social networking sites, they are more likely to believe that their lives are inferior to others when they are confronted with various “groomed” information on social networking sites, which induces upward social comparison [46] and even negative consequences such as a lower self-esteem and lower life satisfaction [42]. The results of previous studies found that upward social comparison negatively predicts self-evaluation and aggravates individuals’ perception of self-differences, and the gap between the actual self and the ideal self stimulates individuals’ desire for material consumption [69]. If consumption behavior can eliminate this self-difference, it can mitigate the negative emotions caused by lower self-evaluation. The self-evaluation maintenance model [50] supports this point from another perspective. When individuals’ self-evaluation is impaired, consumption becomes an important way for individuals to cope with this threat and to maintain and improve their self-awareness. As symbolic consumption, the essence of conspicuous consumption is to strengthen one’s image by showing oneself to others through consumption [3]. By purchasing conspicuous goods, individuals can quickly gain a sense of superiority and build a positive image of themselves, thus compensating for the psychological deficit caused by lower self-evaluation. Previous studies have also shown that the tendency to show off is stronger in upward social comparisons because upward social comparisons are more likely to generate stress and to minimize this psychological stress and express a more dominant self; users are more inclined to engage in showy consumption [70]. The mediated results of upward social comparison also suggest that the higher the intensity of individuals’ use of social networking sites, the more likely it is that upward social comparison will occur and increase the likelihood of conspicuous consumption.

The results of this study suggest that optimism moderates the second half of the path by which the intensity of social networking site usage affects conspicuous consumption through upward social comparison. In other words, the indirect effect is more important for individuals with low optimism levels than for individuals with high optimism levels. This suggests that optimism can influence conspicuous consumption as a moderating variable, which validates research hypothesis H3. This result suggests that positive psychological resources can buffer the tendency toward conspicuous consumption due to upward social comparison, a finding that is consistent with previous results indicating that optimism can play a protective role for individuals [49]. First, based on their low expectations of their own future, college social networking site users with low levels of optimism, when faced with the same upward social comparison information, have the psychological feeling that, regardless of whether they work hard, there is a significant difference between them and the comparison target. As a result, the damage to their self-concept is more serious, so users of social networking sites with low levels of optimism are more prone to psychological perceptions of inferiority. The attributional style theory of optimism suggests that optimism can help individuals make positive attributions in the face of negative events and maintain physical and mental health [71]. Thus, college students with high levels of optimism, based on positive expectations of the future [72], are more inclined to perceive the gap between themselves and others as temporary when comparing themselves to their upwardly mobile counterparts, as a goal state that they can achieve in the future, and to view others’ accomplishments as their possible future achievements [73]; that is, optimism as a positive psychological resource can somewhat attenuate the adverse effects of upward social comparison aftereffects.

In addition, according to psychological compensation theory [18,22], when individuals are unable to solve a need by their original means, they tend to choose other ways to satisfy the need as an alternative. When faced with upward social comparison information, individuals with low optimism levels think that it is difficult to compensate for the enormous difference between themselves and upward comparators through effort and may choose to compensate through other means, such as conspicuous consumption. Compared with acquired efforts, conspicuous consumption can help low-optimism individuals reduce the difference between themselves and others in a more convenient and faster way and close the positive image in the mind of individuals. In contrast, individuals with high levels of optimism may take active steps to try to reduce the differences between themselves and others when they experience pressure from upwardly mobile objects, and the protective effect of optimism weakens the contrast effect of upward social comparison and further weakens individuals’ tendency to consume ostentatiously. Thus, individuals with low levels of optimism are more likely to engage in conspicuous consumption after experiencing upward social comparisons on social networking sites.

This study reveals the impact of the intensity of social networking site usage on conspicuous consumption, the mediating role of upward social comparison, and the moderating role of optimism, which contributes to a deeper understanding of the impact of social networking site usage and the formation mechanism of conspicuous consumption. Meanwhile, this study can help educators correctly view the information presented with positivity bias on social networking sites by reasonably guiding the use of social networking sites and cultivating positive psychological qualities such as optimism to reduce the occurrence of conspicuous consumption.

## 6. Conclusions

In conclusion, our findings suggest that social networking site usage intensity significantly and positively predicts college students’ conspicuous consumption behavior and that upward social comparison partially mediates the relationship between social networking site usage intensity and conspicuous consumption. Furthermore, the second half of this mediating path of intensity of social networking site use–upward social comparison–flaunting consumption was moderated by optimism, i.e., the indirect effect was greater for individuals with low levels of optimism relative to individuals with high levels of optimism.

This study explores the influence of social networking site use on conspicuous consumption through a moderated mediation model, which is not only an extension of previous research on social networking site use and individual consumption behavior but also a deepening of research on social networking site use intensity and conspicuous consumption and a supplement to research on online contextual factors and conspicuous consumption, which helps to deepen the influence mechanism of social networking site use behavior on conspicuous consumption to provide a more moderated mediator model of the influence of social networking site use on individual consumption psychology. The moderated mediation model further deepens the mediation model by explaining both what social networking site use behavior predicts college students’ conspicuous consumption and which individuals’ social networking site use influences conspicuous consumption through upstream social comparison, which improves the explanatory power of the model.

This study has some implications for guiding and promoting college students to establish correct consumption concepts and weakening the negative influence of social networking site use on college students’ conspicuous consumption. First, social networking site usage directly predicts individuals’ conspicuous consumption behavior, so individuals should try to reduce their social networking site usage behavior, such as reducing the length of time spent online and avoiding frequent refreshing of the “Qzone” and “friend circle”. Second, upward social comparison is an important reason why social networking site use behavior affects conspicuous consumption, and college students should try to avoid upward social comparison behavior when using social networking sites, correctly view the “good image” shown by others on social networking sites, and not compare themselves with others arbitrarily to reduce the negative effects of social networking sites. Again, the moderated mediation model and its difference in the level of optimism have some insight in guiding college students to use social networking sites correctly: optimism can significantly alleviate the negative effects of upward social comparison on social networking sites. Therefore, increasing the level of optimistic personality and developing an optimistic mindset is an effective way for individuals to weaken the negative effects of upward social comparisons on social networking sites and thus reduce conspicuous consumption.

Our research has several limitations that provide directions for future research. First, this study is a cross-sectional survey, and the causal relationship between variables may not be completely reliable. In the future, follow-up designs and experimental research can be added for further verification. Second, this study focused on the influence mechanism of SNS use on broadband consumption but did not control for the type of SNS and the main purpose of SNS use and did not consider the two factors of education and economic status in the control of the subjects. These variables may have some influence on the broadband consumption behavior of college students, and the manipulation of these variables can be attempted in future studies to further analyze the influence mechanism of broadband consumption. Third, this study used the SPSS PROCESS macro to test the hypotheses, but this regression method does not separate the errors well, so the structural equation modeling method can be used to test the hypotheses in future studies. Finally, this study uses only college students as the research object to verify the hypothesis, and the results of the study are difficult to generalize to other groups. Therefore, we recommend that future research should be conducted with more groups of subjects.

## Figures and Tables

**Figure 1 behavsci-13-00732-f001:**
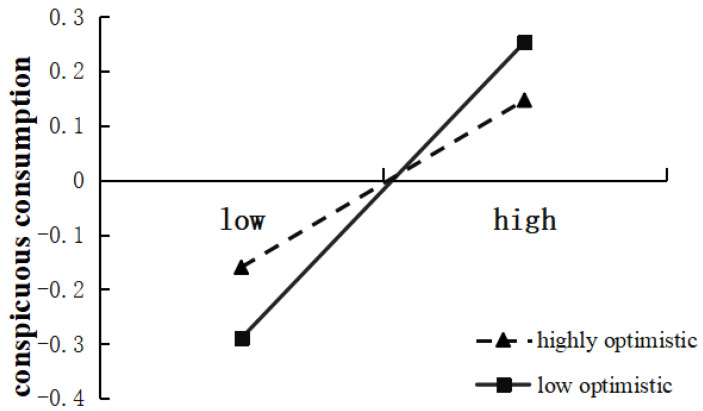
Moderating effect of optimism on the impact of upward social comparison on conspicuous consumption.

**Table 1 behavsci-13-00732-t001:** Correlation matrix of descriptive statistics and coefficients among variables (n = 717).

	*M*	*SD*	1	2	3	4	5	6
1 Gender	0.74	0.44	1					
2 Age	20.08	1.44	−0.11 **	1				
3 Intensity of social network use	0.00	0.63	0.23 **	0.03	1			
4 Conspicuous consumption	2.89	0.66	0.01	0.14 **	0.23 **	1		
5 Upward social comparison	3.00	0.88	0.06	−0.05	0.22 **	0.33 **	1	
6 Optimistic	3.34	0.60	0.11 **	0.02	0.08 *	−0.05	−0.21 **	1

Note: ** *p* < 0.01, * *p* < 0.05.

**Table 2 behavsci-13-00732-t002:** Analysis of intermediary effects (n = 717).

Predictive Variable	Equation (1) (Calibration: Conspicuous Consumption)			Equation (2) (Calibration: Upward Social Comparison)			Equation (3) (Calibration: Upward Social Comparison; Conspicuous Consumption)		
	β	*p*	*t*	β	*p*	*t*	β	*p*	*t*
Gender	−0.02	0.54	−0.61	0.01	0.15	0.33	−0.02	0.46	−0.75
Age	0.09 ***	<0.001	3.64	−0.04	0.75	−1.44	0.10 ***	<0.001	4.27
Social networking site use intensity	0.25 ***	<0.001	6.09	0.26 ***	<0.001	5.28	0.19 ***	<0.001	4.81
Upward social comparison							0.26 ***	<0.001	8.57
*R* ^2^	0.07			0.04			0.16		
	*F*(3, 713) *=* 17.69 ***			*F*(3, 713) *=* 10.72 ***			*F*(4, 712) *=* 32.98 ***		

Note: *** *p* < 0.001.

**Table 3 behavsci-13-00732-t003:** Analysis of moderated mediating effects (n = 717).

Predictive Variable	Equation (1) (Calibration: Conspicuous Consumption)			Equation (2) (Calibration: Upward Social Comparison)			Equation (3) (Calibration: Upward Social Comparison; Conspicuous Consumption)		
	β	*p*	*t*	β	*p*	*t*	β	*p*	*t*
Gender	−0.02	0.54	−0.61	0.01	0.15	0.33	−0.03	0.31	−1.02
Age	0.09 ***	<0.001	3.64	−0.04	0.75	−1.44	0.10 ***	<0.001	4.20
Social networking site use intensity	0.25 ***	<0.001	6.09	0.26 ***	<0.001	5.28	0.19 ***	<0.001	4.80
Optimism							0.01	0.81	0.24
Upward social comparison							0.26 ***	<0.001	8.38
Upward social comparison × optimistic							−0.11 **	0.005	−2.80
*R* ^2^	0.07			0.04			0.17		
	*F*(3, 713) *=* 17.69 ***			*F*(3, 713) *=* 10.72 ***			*F*(6, 710) *=* 23.47 ***		

Note: *** *p* < 0.001, ** *p* < 0.01.

## Data Availability

The data presented in this study are available in Appendix A here.

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
