# Peer review of "The Effect of Social Network Use on Chinese College Students’ Conspicuous Consumption: A Moderated Mediation Model"

_behavsci, 2023, doi:10.3390/bs13090732_

Round 1

Reviewer 1 Report

The paper titled “The effect of social network use on conspicuous consumption: A moderated mediation model” is intriguing, and I appreciate the authors’ efforts. Thank you for the opportunity to review this work. Below are some comments for improvement.

Introduction

·        Clarify the term “Social network use” by providing examples of social networks studied in this research.

·        Expand the definition of “conspicuous consumption” beyond the cited outdated (O'Cass & McEwen, 2004) definition to encompass its current meaning in today’s consumerist society.

Objective and Hypotheses

·        Present the objectives and hypotheses using bullet points for clarity.

·        Support the research hypotheses with appropriate references to enhance their understanding.

Instruments

·        Prove clarity on data collection methods, specifically whether questionnaires were distributed online or offline or both, and if both, how any discrepancies between the two types were addressed for analysis.

·        Describe in detail how participants’ social network use and conspicuous consumption were measured, including the total number of items for each scale.

·        Clearly state the relationship of the Upward social comparison scale and Life orientation test to the measures of students’ social network use, and place this information in the relevant literature session.

Results

·        Offer a comprehensive interpretation of the correlational results obtained.

·        While appreciating the standardization of variables, explain how the authors tested for multicollinearity effects among predictive variables.

·        Define the independent variables, mediators, and dependent variables clearly before presenting the mediation model testing results.

·        Similarly, clearly define “the moderator” in the “Moderated mediation model testing” and provide an interpretation at the end of the results section.

·        Ensure that “the specific impact of regulatory effect” aligns with “the indirect effect of social network use intensity on conspicuous consumption through upward social comparison is moderated by optimism.”

Discussion

·        Ensure that the discussion aligns with the research hypotheses and findings. Highlight which parts of the discussion address each research hypothesis.

·        Provide a clear and concise conclusion that accurately summarizes the study’s findings and aligns with the research objectives.

Overall, the manuscript holds significant value for readers. However, addressing the above comments will improve the overall quality of the paper.

Author Response

Revised as suggested by the first reviewer of the manuscript:

The paper titled “The effect of social network use on conspicuous consumption: A moderated mediation model” is intriguing, and I appreciate the authors’ efforts. Thank you for the opportunity to review this work. Below are some comments for improvement.

  1. Introduction
  • Clarify the term “Social network use” by providing examples of social networks studied in this research.

A: Thanks to the reviewers for their comments. The expression on the use of the term social networking sites is indeed insufficient, and according to the reviewers' suggestions, corresponding modifications and additions have been made in the introduction section (see pp. 3-4), as follows:

Social network sites (SNSs) refer to network services based on a bounded system, such as WeChat, Weibo, and QQ. Through public or semipublic personal pages, users are able to connect with each other and to view and forward their own links as well as those of other users, as well as to use social networking sites to present themselves or to develop and maintain relationships with others (Boyd & Ellision, 2008; Brailovskaia & Bierhoff, 2020). Due to their fast information dissemination and wide range of dissemination, social networking sites have become an important platform for interpersonal communication in the information age.

  • Expand the definition of “conspicuous consumption” beyond the cited outdated (O'Cass & McEwen, 2004) definition to encompass its current meaning in today’s consumerist society.

A: Thank you to the reviewers for their comments. In response to your comments, the definition of "conspicuous consumption" has been expanded in the text (see pp. 2) as follows:

Conspicuous consumption refers to the behavior of individuals who reinforce their image by consuming openly or showing their status to others (O'Cass & McEwen, 2004). It is specifically manifested when individuals seek the nonfunctional value of goods to purchase goods or services that are symbolic and able to demonstrate their status or build their self-image to others (Roy Chaudhuri et al., 2011).

  1. Objective and Hypotheses
  • Present the objectives and hypotheses using bullet points for clarity.

A: Thanks to the reviewers for their comments. The bullet points have been used in the text to present the research objectives and hypotheses more clearly.

  • Support the research hypotheses with appropriate references to enhance their understanding.

A: We thank the reviewers for their suggestions, and we have incorporated additional literature and combed some of the literature review with comments from other reviewers.

  1. Instruments
  • Prove clarity on data collection methods, specifically whether questionnaires were distributed online or offline or both, and if both, how any discrepancies between the two types were addressed for analysis.

A: Thank you to the reviewers for their comments. In this study, a cluster random sampling method was used to collectively administer the test and collect data through an online survey using a web-based questionnaire.

  • Describe in detail how participants’social network use and conspicuous consumption were measured, including the total number of items for each scale.

A: Thanks to the reviewers for their comments.

This study used the Social Networking Site Intensity of Use Questionnaire to measure the intensity of social network use. The questionnaire has 8 items. The first 2 items measure the number of friends and average daily use time of individuals on social networking sites, and the last 6 items use a 5-point Likert scale (1 "very little" ~ 5 "very much") to measure the strength of the emotional connection between social networking sites and individuals and the degree of penetration into the individual's life. After converting the individual scores into standardized scores, the total mean score was calculated, with higher scores indicating a higher intensity of SNS use.

In this study, the Conspicuous Consumption Scale (CCS) compiled by Marcoux (Marcoux et al., 1997) and revised by Chen Xu (2009) was used to measure conspicuous consumption. The questionnaire consists of 13 items and measures the individual's tendency to consume ostentatiously by means of a 5-point Likert scale (1 "very inconsistent" to 5 "very consistent"); the higher the score is, the stronger the tendency to consume ostentatiously.

  • Clearly, state the relationship of the Upward social comparison scale and Life orientation test to the measures of students’social network use, and place this information in the relevant literature session.

A: We thank the reviewers for their comments. We have strengthened the discussion of relevant variables in the text (see pp. 4) and added new references.

  1. Results
  • Offer a comprehensive interpretation of the correlational results obtained.

A: We thank the reviewers for their comments. We have further discussed the relevant results obtained and revised them in the text (see pp. 15-16).

  • While appreciating the standardization of variables, explain how the authors tested for multicollinearity effects among predictive variables.

A: Thanks to the reviewers for their comments. In this paper, the VIF (Variance Inflation Factor) test is used. 1/(1 - Rj2) is defined as the variance inflation factor of the jth variable, where Rj2 denotes the coefficient of determination obtained by fitting the regression equation of the jth variable as the dependent variable with the remaining independent variables, and the larger the value of the VIF is, the stronger the correlation between the variable and the remaining independent variables. In this paper, the variance inflation factors (VIFs) of SNS use, upward social comparison, optimism and conspicuous consumption are not higher than 5, indicating that there is no multicollinearity problem.

  • Define the independent variables, mediators, and dependent variables clearly before presenting the mediation model testing results.

A: Thanks to the reviewers for their comments. The independent, mediating and dependent variables have been clearly defined in the results of the mediation model test and modified in the text (see pp. 10) as follows:

After standardizing the independent variable social networking site use, the dependent variable conspicuous consumption, and the mediating variable upward social comparison, the mediating role of upward social comparison in the relationship between social networking site use and conspicuous consumption was analyzed using Hayes's SPSS macro program PROCESS.

  • Similarly, clearly define “the moderator”in the “Moderated mediation model testing”and provide an interpretation at the end of the results section.

A: Thanks to the reviewers for their comments. The "moderator" has been clearly defined in the "Test of Moderated Mediation Model" and modified in the text (see pp. 11) as follows:

Model 14 was used to test the moderating effect of optimism, analyzing whether the mediating role of upward social comparison between social networking site use and conspicuous consumption (second half) is moderated by optimism.

  • Ensure that “the specific impact of regulatory effect”aligns with “the indirect effect of social network use intensity on conspicuous consumption through upward social comparison is moderated by optimism.”

A: Thanks to the reviewers for their comments. It has been confirmed that "the specific impact of regulatory effects" is consistent with "the indirect effect of social network usage intensity on conspicuous consumption is optimistically moderated by upward social comparisons".

  1. Discussion
  • Ensure that the discussion aligns with the research hypotheses and findings. Highlight which parts of the discussion address each research hypothesis.

A: Thank you for the reviewer's comments. Based on your suggestions, we have revised the discussion section in the text (see pp. 15-16).

  • Provide a clear and concise conclusion that accurately summarizes the study’s findings and aligns with the research objectives.

A: Thank you for the reviewer's comments. According to your suggestions, we summarize the findings and add them accordingly in the paper (see PP.18) as follows:

(1) Intensity of SNS use significantly and positively predicts college students' conspicuous consumption behavior.

(2) Upward social comparison mediated the relationship between SNS usage intensity and conspicuous consumption.

(3) Optimism negatively moderated the mediating effect of upward social comparison between SNS use intensity and conspicuous consumption.

Overall, the manuscript holds significant value for readers. However, addressing the above comments will improve the overall quality of the paper.

Reviewer 2 Report

This is an interesting and novel topic that needs more attention in the academia.  

-The research gap is clear. (E.g., However, there is a relative lack of research on the factors influencing online contexts, and only a small number of scholars have studied and confirmed the positive predictive effect of social networking site use on conspicuous consumption (Marcoux et al.,1997; Podoshen & Andrzejewski, 2012; Thoumrungroje, 2014; Wilcox & Stephen, 2013).

-The hypotheses are logical and built on the theoretical framework (e.g., Veblen, 2019) 

-Up-to-date research has been cited (e.g., Midgley et al., 2021). 

-Methodology part is clearly described the procedure of the study (e.g., The experiment procedure, measurement scale, the result of CFA) . 

-An appropriate statistical analyses were used to test the mediated moderation and mediated model’s hypotheses (e.g., PROCESS Model 4 & 14). 

-Results are clear. 

-Implications and limitations sections are clear.  

Author Response

Revised as suggested by the second reviewer of the manuscript:

## reviewer 2

This is an interesting and novel topic that needs more attention in academia.

-The research gap is clear. However, there is a relative lack of research on the factors influencing online contexts, and only a small number of scholars have studied and confirmed the positive predictive effect of SNS use on conspicuous consumption (Marcoux et al.,1997; Podoshen & Andrzejewski, 2012; Thoumrungroje, 2014; Wilcox & Stephen, 2013).

-The hypotheses are logical and built on the theoretical framework (e.g., Veblen, 2019)

-Up-to-date research has been cited (e.g., Midgley et al., 2021).

-Methodology part is clearly described the procedure of the study (e.g., The experiment procedure, measurement scale, the result of CFA).

-Appropriate statistical analyses were used to test the mediated moderation and mediated model’s hypotheses (e.g., PROCESS Model 4 & 14).

-Results are clear.

-Implications and limitations sections are clear.

A: Thank you to the reviewer.

Reviewer 3 Report

Dear authors,

It is a very interesting study and research topic. However, I would suggest improving the paper structure by adding two additional sections: literature review and conclusion. This way, your paper would have six sections instead of four. Why am I suggesting this? First, the paper will be more logically organised. Additionally, this way, you would avoid having too long sections. For example, the introduction is really long, including the literature review and hypothesis development.

Additionally, I would suggest expanding the list of references, especially when you cite slightly dated studies. Search for newer studies in the same field and add them next to existing references in your paper.

Finally, I would suggest referring to the sample in the article title. Since the sample compositions contribute to the study's limitation, I believe it should be addressed in the title. On top, the sample (and the topic) has a specific geographic context. Again, it is something that could be addressed in the article title.

Author Response

Revised as suggested by the third reviewer of the manuscript:

## 3

Dear authors,

It is a very interesting study and research topic. However, I would suggest improving the paper structure by adding two additional sections: literature review and conclusion. This way, your paper would have six sections instead of four. Why am I suggesting this? First, the paper will be more logically organized. Additionally, this way, you would avoid having too long sections. For example, the introduction is truly long, including the literature review and hypothesis development.

A: Thanks to the reviewer. We have added two sections, the literature review and the conclusion, to improve the structure of the paper. We have also added more literature and combed some of the literature review with comments from other reviewers.

Additionally, I would suggest expanding the list of references, especially when you cite slightly dated studies. Search for newer studies in the same field and add them next to existing references in your paper.

A: Thanks to the reviewer. We have incorporated some new literature with comments from other reviewers and added it to the references section.

Finally, I would suggest referring to the sample in the article title. Since the sample compositions contribute to the study's limitation, I believe it should be addressed in the title. In addition, the sample (and the topic) has a specific geographic context. Again, it is something that could be addressed in the article title.

A: Thank you, reviewer. According to your suggestion, we have added "Chinese college students" to the title, and the revised title is "The impact of social network usage on Chinese college students' conspicuous consumption: a moderated mediation model".

Round 2

Reviewer 1 Report

I have reviewed the authors’ replies to my feedback. They have addressed all of my comments effectively. I don’t have any further remarks to make. In my opinion, the manuscript is ready for publication.

Author Response

A: Thank you to the reviewer for recognizing.

Reviewer 3 Report

I appreciate all the updates and structural changes you made so far. There is only one more thing I would suggest you improve - 6. Conclusion. It should be more elaborated. I do not find it appropriate in the current form (i.e. a numbered list). Elaborate it in more details. Transfer research limitations from 5. Discussion to 6. Conclusion. Also, implications and guidelines for further research should find their place in the conclusion section of the paper.

Author Response

Revised as suggested by the third reviewer of the manuscript:

I appreciate all the updates and structural changes you made thus far. There is only one more thing I would suggest you improve - 6. Conclusion. It should be more elaborated. I do not find it appropriate in the current form (i.e., a numbered list). Elaborate it in more details. Transfer research limitations from 5. Discussion to 6. Conclusion. Additionally, implications and guidelines for further research should find their place in the conclusion section of the paper.

A: We thank the reviewer for your comments. Based on your comments, we have revised the conclusion section of the text and added the limitations of the research from the discussion section to the conclusion section (see pp. 11-12), as follows:

  1. Conclusion

In conclusion, our findings suggest that social networking site usage intensity significantly and positively predicts college students' conspicuous consumption behavior and that upward social comparison partially mediates the relationship between social networking site usage intensity and conspicuous consumption. Furthermore, the second half of this mediating path of intensity of social networking site use-upward social comparison-flaunting consumption was moderated by optimism, i.e., the indirect effect was greater for individuals with low levels of optimism relative to individuals with high levels of optimism.

This study explores the influence of social networking site use on conspicuous consumption through a moderated mediation model, which is not only an extension of previous research on social networking site use and individual consumption behavior but also a deepening of research on social networking site use intensity and conspicuous consumption and a supplement to research on online contextual factors and conspicuous consumption, which helps to deepen the influence mechanism of social networking site use behavior on conspicuous consumption to provide a more moderated mediator model of the influence of social networking site use on individual consumption psychology. The moderated mediation model further deepens the mediation model by explaining both what social networking site use behavior predicts college students' conspicuous consumption and which individuals’ social networking site use influences conspicuous consumption through upstream social comparison, which improves the explanatory power of the model.

This study has some implications for guiding and promoting college students to establish correct consumption concepts and weakening the negative influence of social networking site use on college students' conspicuous consumption. First, social networking site usage directly predicts individuals' conspicuous consumption behavior, so individuals should try to reduce their social networking site usage behavior, such as reducing the length of time spent online and avoiding frequent refreshing of the "Qzone" and "friend circle". Second, upward social comparison is an important reason why social networking site use behavior affects conspicuous consumption, and college students should try to avoid upward social comparison behavior when using social networking sites, correctly view the "good image" shown by others on social networking sites, and not compare themselves with others arbitrarily to reduce the negative effects of social networking sites. Again, the moderated mediation model and its difference in the level of optimism have some insight to guide college students to use social networking sites correctly: Optimism can significantly alleviate the negative effects of upward social comparison on social networking sites. Therefore, increasing the level of optimistic personality and developing an optimistic mindset is an effective way for individuals to weaken the negative effects of upward social comparisons on social networking sites and thus reduce conspicuous consumption.

Our research has several limitations that provide directions for future research. First, this study is a cross-sectional survey, and the causal relationship between variables may not be completely reliable. In the future, follow-up designs and experimental research can be added for further verification. Second, this study focused on the influence mechanism of SNS use on broadband consumption but did not control for the type of SNS and the main purpose of SNS use and did not consider the two factors of education and economic status in the control of the subjects. These variables may have some influence on the broadband consumption behavior of college students, and manipulation of these variables can be attempted in future studies to further analyze the influence mechanism of broadband consumption. Third, this study used the SPSS PROCESS macro to test the hypotheses, but this regression method does not separate the errors well, so the structural equation modeling method can be used to test the hypotheses in future studies. Finally, this study uses only college students as the research object to verify the hypothesis, and the results of the study are difficult to generalize to other groups. Therefore, we recommend that future research should be conducted with more groups of subjects.